# Characterization, Antioxidant, Anti-Aging and Organ Protective Effects of Sulfated Polysaccharides from *Flammulina velutipes*

**DOI:** 10.3390/molecules24193517

**Published:** 2019-09-28

**Authors:** Fangfang Yuan, Zheng Gao, Wenbo Liu, Huaping Li, Yiwen Zhang, Yanbo Feng, Xinling Song, Wenshuai Wang, Jianjun Zhang, Chunyan Huang, Le Jia

**Affiliations:** 1Institute of Agricultural Resources and Environment, Shandong Academy of Agricultural Science, Key Laboratory of Wastes Matrix Utilization, Ministry of Agriculture, Jinan 250100, China; 18815388639@163.com; 2College of Life Science, Shangdong Agricultural University, Taian 271018, China; gaoguo000@sina.cn (Z.G.); 18706389772@163.com (W.L.); 18854890971@163.com (H.L.); zhangyiwenwa@126.com (Y.Z.); fungyimbo@163.com (Y.F.); sxl19930311@163.com (X.S.); wws0903as@163.com (W.W.); yyxf0315@163.com (J.Z.)

**Keywords:** sulfated polysaccharides, *Flammulina velutipes*, antioxidation, anti-aging effects, organ protection

## Abstract

As an irreversible and complex degenerative physiological process, the treatment for aging seems strategically necessary, and polysaccharides play important roles against aging owing to their abundant bioactivities. In this paper, the antioxidant and anti-aging activities of *Flammulina velutipes* polysaccharides (FPS) and its sulfated FPS (SFPS) on d-galactose-induced aging mice were investigated. The in vitro antioxidant activities demonstrated that SFPS had strong reducing power and superior scavenging effects on 2, 2-diphenylpicrylhydrazyl (DPPH), hydroxyl radicals and the chelating activities of Fe^2+^. The in vivo animal experiments manifested that the SFPS showed superior antioxidant and protective abilities against the d-galactose-induced aging by increasing the antioxidant enzyme activities, decreasing lipid peroxidation, improving the inflammatory response and ameliorating the anile condition of mice. Furthermore, the structural analysis of SFPS was investigated through FT-IR, NMR, and HPLC analysis, and the results indicated that SFPS was a homogeneous heteropolysaccharide with a weight-average molecular weight of 2.81 × 10^3^ Da. Furthermore, SFPS has also changed in characteristic functional groups and monosaccharide composition compared to FPS. These results suggested that sulfated modification could enhance the anti-oxidation, anti-aging and protective activities of *F. velutipes* polysaccharides, which may provide references for the development of functional foods and natural medicines.

## 1. Introduction

It has been reported that aging is an irreversible and complex degenerative physiological process that occurs in various parts of the body and can lead to the gradual loss of body function and increases the risk of various chronic diseases such as hypertension, diabetes, cognitive impairment, atherosclerosis, Alzheimer’s disease (AD) and multiple cancers [1,2,3]. The cholinergic hypothesis is one of the main mechanisms of AD, and acetylcholine (Ach) is an important neurotransmitter in the body, which plays an important role in learning and memory. The d-galactose aging animal model has the advantages of simple and easy modeling, low prices and stable results [4]. It is the most commonly used method for preparing aging animal models in the world. This model is widely used in the screening of drugs for delaying aging and preventing and treating Alzheimer’s disease [5,6]. Studies have shown that the long-term injection of d-galactose solution can produce excessive reactive oxygen species (ROS), which reduces the activities of antioxidant enzymes in various organs, and forms more superoxide anions and various oxidation products, and causes cell damage, resulting in a decreased function of multiple organs and multiple systems in the body [7]. However, many experiments have confirmed that d-galactose could cause degenerative changes like natural aging in many tissues [8]. Free radical theory is currently considered to be the most reasonable mechanism for inducing aging [9]. The excessive production of reactive oxygen species (ROS) can lead to oxidative stress and pro-inflammatory reactions, which cause redox disequilibrium, inducing dangerous effects of cellular proteins, lipids and DNA, thereby inhibiting normal cell function and even leading to apoptosis [10,11]. Fortunately, antioxidants play an important role in oxidative stress, which can react with free radicals to remove them, slow down the aging process, and reduce the risk of disease. However, synthetic antioxidants are suspected of being toxic to the body during long-term use, and natural sources of antioxidants have received much attention for their safety, efficacy and potential beneficial biological activities [12,13]. Based on this, it is a new research interest to find natural ingredients with anti-oxidation, immune regulation and anti-inflammatory properties to prevent aging.

In recent years, edible mushrooms have become a kind of food with high dietary and nutritional value because of its delicious taste, rich nutrition and including various biological active substances such as polysaccharides, vitamins, proteins, polyphenolic compounds, flavonoids and carotenoids [14]. Edible mushroom polysaccharide is an active polysaccharide that controls cell division and differentiation, regulates cell growth and senescence. It has been reported that polysaccharides can scavenge free radicals and play an important role in preventing the body’s oxidative damage and maintaining the organic metabolic balance [15]. *Flammulina velutipes* is one of the most common economic edible fungi. Reports have shown that polysaccharides from *F. velutipes* fruiting bodies have good immune regulation, anti-oxidation, anti-aging, cholesterol lowering, anti-tumor and other physiological activities [16,17,18]. These activities suggested that *F. velutipes* is precious in pharmacology and the health care industry. However, no report about the aging effects of sulfated modified polysaccharides by *F. velutipes* against d-galactose induced aging mice has been published to date.

In present experiment, the in vitro antioxidant activities, antiaging and protective effects of *F. velutipes* polysaccharides (FPS) and its sulfated *F. velutipes* (SFPS) in vivo were investigated, as well as the structural properties, aiming to provide references for clinical treatment of aging and its complications in the d-galactose induced aging mice.

## 2. Results

### 2.1. Structural Characterization

#### 2.1.1. The Purity and Molecular Parameters

The purity of SFPS and FPS were analyzed by UV spectrophotometer. The results are shown in Figure 1. No obvious absorptions were observed in 260 and 280 nm, indicating that there was no protein and nucleic acid in SFPS and FPS. HPLC indicated that the weight-average molecular weight (Mw), number-average molecular weight (Mn) and z-average molecular weight (Mz) of SFPS were 2.81 × 10^3^, 1.47 × 10^3^ and 7.13 × 10^3^ Da, while the Mw, Mn and Mz of FPS were 6.96 × 10^6^, 6.25 × 10^6^ and 7.67 × 10^6^ Da, respectively. Both SFPS and FPS exhibited narrow and homogeneous properties on HPLC, indicating that SFPS and FPS are homogeneous polysaccharides.

#### 2.1.2. FT-IR Assay

As shown in Figure 2, the FT-IR spectra of all samples were similar indicating that the basic skeleton of the FPS remains unchanged after sulfation.

The properties of SFPS and FPS showed typical and strong absorption peaks at 3427 cm^−1^ and 3435 cm^−1^ for the -OH stretching vibrations, and weak absorption bands at approximately 2900 cm^−1^ C-H stretching vibrations. Kizil et al. showed the typical characteristics of polysaccharides [19]. The strong absorption at 1637 cm^−1^ and 1644 cm^−1^, corresponding to the stretching vibration of the C=O, and the peak at approximately 1380 cm^−1^ represented the angular deformation of C-H (CH_3_ group). The bands at approximately 1050 cm^−1^ indicated the presence of pyranose [20]. Compared to SFPS, the absorption of FPS at 1542 cm^−1^ and 1488 cm^−1^, corresponding to the bending vibration of the N-H bond, showed the existence of protein. The bands at approximately 850 cm^−1^ and 900 cm^−1^ might be ascribed to the α- and β-glucosyl linkage [21]. The SFPS had a strong absorption peak at 1241 cm^−1^, which was caused by asymmetric S=O stretching vibration absorption. In addition, the absorption peak at 805 cm^−1^ was caused by the absorption of tensile vibration of C-O-S [22]. After sulfation, the S content of the sulfated derivatives reached 0.38. It seemed that the sulfated modification was successful.

#### 2.1.3. NMR Analysis

The ^1^H and ^13^C NMR spectra of FPS and SFPS can be observed in Figure 3A–D. The ^1^H NMR spectrum of FPS contained six signals at 5.16, 5.02, 4.97, 4.91, 4.70, and 4.43 ppm corresponding to the anomeric protons (Figure 3A). While the ^1^H NMR spectrum of SFPS emerged a set of wide and intense signals (3.0–4.0 ppm), indicating that CH_2_-O and CH-O groups were existed in SFPS [23]. In the anomeric region of ^13^C NMR spectrum showed that the FPS represented two anomeric carbons at 103.17 and 97.82 ppm, indicating that FPS contained both α- and β-anomeric configurations. The ^13^C NMR spectroscopy of SFPS (Figure 3D) had the signal at 102.57, 96.36 and 95.81 ppm, corresponding to α- and β-glucosyl [24]. Moreover, the important signals from 60 to 80 ppm were characteristic of carbohydrates. A signal at a low field from 20 to 40 ppm was also found, which suggested a characteristic glucan–protein compound structure. The signal at approximately 69 ppm was due to C-6, and showed the presence of α-1-glucoside bonds and α-1, 6-glucoside bonds [25].

#### 2.1.4. Monosaccharide Composition Analysis

The monosaccharide compositions of SFPS and FPS were identified by comparing the retention time with those of the standards. In present work, the monosaccharide compositions analysis (Figure 4B) showed that FPS mainly contained nine monosaccharides including mannose, ribose, rhamnose, glucuronic acid, glucosamine, glucose, galactose, xylose, and fucose with a percentage composition of 13.95%, 1.98%, 0.97%, 0.82%, 0.22%, 53.35%, 20.08%, 3.24% and 5.39%, the SFPS mainly contained six monosaccharides including mannose, ribose, glucose, galactose, xylose, and fucose with the percentage of 9.27%, 4.47%, 47.55%, 29.98%, 3.78% and 4.93% (Figure 4C). These data suggested that the number of monosaccharide constituents of SFPS was reduced compared with FPS, but the galactose levels were significantly improved. The content of galactose in SFPS was agreement with the antioxidant activities, indicating galactose may play vital roles in maintaining the antioxidant status.

### 2.2. Antioxidant Capacities In Vitro

In this experiment, four indicators including scavenging effects on hydroxyl, superoxide and DPPH radicals and reducing power, were used to determine the antioxidant function of SFPS and FPS in vitro (Figure 5). The results appeared in a dose-dependent linear correlation. At the concentration of 1000 mg/L, the scavenging abilities of SFPS on hydroxyl, superoxide, and DPPH radicals reached 63.43 ± 2.25%, 68.58 ± 2.42%, and 56.82 ± 2.21%, which were 22.59%, 60.68%, and 45.99% higher than that of FPS respectively (Figure 5A–C). As can be seen from Figure 5D, the reducing power of SFPS or FPS reached 0.60 ± 0.02 or 0.29 ± 0.02 at the concentration of 1000 mg/L, while the value of SFPS was 106.90% higher than that of FPS, indicating SFPS had higher reducing power ability and stronger antioxidant capacity in vitro.

### 2.3. Acute and Subchronic Toxicity Analysis

During the gavage for 14 days, no abnormalities in their eating, drinking, activity, mental state, smooth fur, normal feces, and no death were observed. There was no abnormality in the mental state, activity status, drinking water, and fecal urine of the mice in each dose group. The mice in the 6 dose groups (SFPS/FPS) were not significantly different from the normal control (NC) group in weight, hematological parameters and serum biochemical indexes after 28 days of continuous gavage, and male mice showed superior tolerance. The results demonstrated that the polysaccharides of SFPS and FPS showed superior biocompatibilies.

### 2.4. Anti-Aging Capacities In Vivo

#### 2.4.1. Body Weight and Organ Index

As shown in Table 1, after the eighth week, the d-gal model group had a significant decrease in body weight compared with the control group (*p* < 0.01), and the remaining administration groups reversed d-gal-induced weight loss (*p* < 0.05). Obviously, the brain index, liver index, and kidney index of the model group decreased compared with the blank control group (*p* < 0.05), while the administration of SFPS or FPS could increase the brain index, liver index, and kidney index compared to the d-gal- induced aging group (*p* < 0.05).

#### 2.4.2. Effects of SFPS and FPS on Alanine Aminotransferase (ALT), Aspartate Aminotransferase (AST), Alkaline Phosphatase (ALP), Creatinine Levels (CRE), Blood Urea Nitrogen Levels (BUN), Uric Acid (UA), High Density Lipoprotein Cholesterol (HDL-C) and Low-Density Lipoprotein Cholesterol (LDL-C)

As shown in Figure 6, the ALT, AST and ALP activities of mice after the d-gal injection significantly increased compared to the NC group (*p* < 0.01), suggesting that characteristic liver damages had occurred (Figure 6A–C). Fortunately, these enhancements could be inhibited by the treatment with SFPS (*p* < 0.05) or FPS (*p* < 0.05). In particular, SFPS at the dosage of 400 mg/kg, the ALT, AST and ALP activities decreased to 55.23 ± 3.57, 135.34 ± 4.90 and 118.43 ± 5.11 U/L, respectively. Furthermore, the kidney index in the MC group significantly increased (*p* < 0.01) when compared to the NC groups, indicating that kidneys were severely damaged by the d-gal-injection. Simultaneously, the CRE, BUN, and UA levels reduced to 68.31 ± 3.13 μmol/L, 6.92 ± 0.30 and 312 ± 13.25 mmol/L with the decreasing rates of 33.33, 32.35 and 36.97% by the treatment with SFPS at the dose of 400 mg/kg, which were lower than the MC groups, respectively. As described in Figure 6G,H, when compared with the NC group, the mice of the MC group showed a significant decrease of HDL-C levels, a visible increase of LDL-C levels, which demonstrated that serious lipid metabolism disordered occurred after the d-gal injection. Obviously, for mice in the high dose groups treated with SFPS, the HDL-C levels reached 2.38 ± 0.09 mmol/L, which was higher than that in the FPS-treated mice (2.05 ± 0.08 mmol/L), as well as the LDL-C reached 0.53 ± 0.02 mmol/L, which was lower than that in the FPS-treated mice, indicating that SFPS had the effect of improving lipid metabolism.

#### 2.4.3. Effects of SFPS and FPS on Antioxidant Status

In order to better understand the association between aging induced by d-galactose and oxidative stress and organ damage, the enzymatic activities, non-enzymatic activities and lipid peroxidation were determined. The results are shown in Figure 7. Obviously, the superoxide dismutase (SOD), glutathione peroxidase (GSH-Px), catalase (CAT) and the total antioxidant capacity (T-AOC) activities of mice in the MC group were significantly (*p* < 0.01) lower, while the malondiadehyde (MDA) and lipid peroxide (LPO) contents were remarkably higher than that in the NC group, respectively, suggesting that severe oxidative stress had been induced by d-galactose. In this work, the activities of SOD, GSH-Px, CAT, and T-AOC were expressed in a positive dose-dependent manner by the treatment with SFPS or FPS at the detected doses of 200 or 400 mg/kg. The liver SOD, GSH-Px, CAT, and T-AOC activities of SFPS group reached to 132.49 ± 4.87, 124.46 ± 4.38, 209.72 ± 6.63 and 58.63 ± 2.57 U/mg prot at the high dose group, increased significantly (*p* < 0.01) compared with those of the model group (95.04 ± 4.85, 80.37 ± 3.62, 112.63 ± 6.65 and 33.69 ± 2.26 U/mg prot) (Figure 7A–D). As displayed in Figure 7E,F, when compared with that in the NC group, the contents of MDA and LPO significantly increased (*p* < 0.01) in the d-galactose-induced aging mice. The LPO and MDA contents of mice treated with SFPS at 400 mg/kg reached 7.87 ± 0.51 nmol/mg prot and 7.57 ± 0.46 μmol/mg prot, were 30.37% and 42.54% lower than that in the FPS group, nearest to those of the normal group. These results demonstrated that SFPS could resist d-galactose-induced aging by increasing the activities of SOD, GSH-Px, CAT, and T-AOC and reducing the contents of MDA and LPO.

#### 2.4.4. Effects of SFPS and FPS on Acetylcholin Esterase (Ach E) and Nitric Oxide Synthase (NOS) in the Brain

Clinically, the activities of Ach E and NOS in the brain could be used to detect biochemical indicators of aging. As illustrated in Figure 8, the activities of Ach E in the brain expressed a significant increase in the MC group when compared with the NC group (*p* < 0.01), while the NOS activities were markedly decreased, indicating that the aging damage had occurred in the brain. Fortunately, after treatment with SFPS or FPS by intragastric administration, there was an obvious change in the activities of Ach E and NOS. The Ach E activities of the SFPS group at 400 mg/kg were 0.48 ± 0.025 U/mg prot, which was 35.42% lower than those of the MC group, while the NOS value reached 1.31 ± 0.062 μ/mL with 28.43% higher than the MC group, supporting that SFPS and FPS intended to restore d-gal- induced aging impairments by increasing NOS activities (*p* < 0.05) and inhibiting Ach E activities.

### 2.5. Histopathological Analysis

In the current work, the histopathological observation of the liver, kidney and brain were performed by an optical microscope (400×). The results are shown in Figure 9. Obviously, the organ damage was mainly evidenced by the cellular and nucleus degradations. When compared to the normal group of hepatocytes, the morphological destruction of liver tissue in aging mice were reflected by serious balloon degeneration, lipid droplet accumulation, cytoplasmic loosening, necrosis and inflammatory infiltration (Figure 9A). The severe kidney damage of glomerulus destruction, tubularointerstitial lesions, glomerular sclerosis, vacuolation of tubular epithelial cells and loss of brush borders (Figure 9B) were observed in the MC group. However, in the normal group, the morphology and structure of neurons in the cortex (Figure 9C) and hippocampus region (Figure 9D) remained with integrity. The cytoplasm and nucleus were centered with clear staining and the neurons remained well-aligned. The mice in the MC groups showed necrotic cell infiltration, loosened cytoplasm and vacuolar neural fiber were arranged confusingly and disorderly. Interestingly, the fed with SFPS or FPS at four dosage groups were significantly ameliorated with organ damage at different levels. In particular, with SFPS at the dosage of 400 mg/kg, the repair effect was close to the blank control group.

## 3. Discussion

In recent years, mushroom polysaccharides have been proven to have high antioxidant activity and assisted in the prevention of oxidative damage induced by ROS in living organisms [26]. Furthermore, previous literature had demonstrated that dysfunction of the brain and liver is closely related to aging and is susceptible to d-gal caused by oxidative injury [27,28]. The accumulation of senescent cells played an important role in various disease tissues, and its reduction of senescent cells could delay the damage of tissue function and prolong life [29]. It has been reported that *F. velutipes* polysaccharides showed superior antioxidant activities both in vitro and in vivo [30,31]. However, there are a few reports about sulfated modified polysaccharides. Therefore, it is quite significant to investigate the protective effects of SFPS in d-galactose-induced aging mice on liver and brain injuries. It has been reported in the literature that hydroxyl radicals are the main reactive oxygen species in biological tissues, which induce DNA base oxidation and glycoside cleavage, causing permanent damage to DNA, leading to metabolic abnormalities, thereby inducing various diseases [32]. Meanwhile, superoxide anions were generally considered to be precursors of other free radicals, such as hydroxyl radicals, which are the cause of many diseases. Therefore, the removal of superoxide anion radicals is essential for the protection of living systems [33]. Experimentally, the DPPH free radicals are synthetic nitrogen-centered stable free radicals. The method of scavenging DPPH free radicals has been widely used for evaluating the antioxidant capacity of natural compounds and medicine. In addition, reducing power is an important indicator of antioxidant capacity of biological compounds. By providing a hydrogen atom to destroy the free radical chain, the free radical becomes a stable substance, therefore interrupting the radical chain and preventing the formation of peroxidation products [34]. Therefore, the antioxidant properties of polysaccharides are related to their abilities of free radical scavenging and reducing capacities [35]. In our present work, SFPS showed stronger scavenging effects than FPS on hydroxyl, superoxide anions and DPPH radicals, as well as superior reducing power, demonstrating that the SFPS had potential antioxidant effects in vitro.

It was well-known that the AST, ALT and ALP activities in serum were clinically considered as biochemical markers for monitoring liver damage. These can enter the bloodstream from hepatocytes when the permeability of the cell membrane is impaired. Serum enzyme activities were significantly elevated when the liver was damaged [36]. Clinically speaking, the abnormal elevated levels of CRE, BUN and UA were usually used for a kidney examination, which was the premonition of severe finally uremia. BUN, as the first endogenous substance produced by proteolysis, was excreted by the filtration of the glomerulus. CRE was a byproduct of creatine and phosphocreatine catabolism. It was endogenously produced and released into body fluids, which was an indicator of glomerular filtration rate [37]. UA was also an index parameter of renal dysfunction. In this work, the treatment of SFPS could have receded the activities of ALT, AST and ALP, as well as decreased the CRE, BUN and UA levels, indicating that SFPS had potential protective effects in d-gal-induced aging mice. In addition, the elevated levels of LDL-C and reduced levels of HDL-C are major causes of liver disease. Therefore, improving lipid metabolism is essential in the prevention and treatment of cardiovascular diseases. Apparently, the present results showed that SFPS significantly increased the HDL-C levels and decreased LDL-C levels, suggesting that the polysaccharides from *F. velutipes* had the potential abilities in repairing the d-gal induced aging liver injury.

According to the previous research, the oxidative damage of the liver may be caused by free radicals produced by lipid peroxidation. The aging body produces a large amount of ROS, which reduces the body’s ability to scavenge free radicals and the activity of antioxidant enzymes, causing oxidative damage to the body [38]. In order to protect cells from oxidative stress, the body has formed various defense systems such as SOD, GSH-Px, CAT and T-AOC. The SOD is a naturally occurring superoxide free radical scavenger that converts harmful superoxide radicals into hydrogen peroxide [39]. The GSH-Px reduces the toxic peroxide to a non-toxic hydroxy compound while promoting the decomposition of hydrogen peroxide, and the main role of CAT is to catalyze the decomposition of H_2_O_2_ into H_2_O and O_2_, and to remove hydrogen peroxide from the body, thus protecting the structure and function of the cell membrane from peroxide damage [40]. Moreover, the T-AOC activities could reflect or characterize the body’s ability of non-enzymatic antioxidant defense systems [41]. In addition, MDA and LPO are lipid-peroxidation products produced by lipid peroxidation, which are associated with aging, cell destruction and the occurrence of certain diseases [42]. In this work, compared with the normal group, the activity of SOD, GSH-Px, CAT and T-AOC in the model group significantly decreased while the contents of MDA and LPO observably increased, indicating that severe oxidative damage had occurred. Interestingly, after the treatment with SFPS, the activity of antioxidant enzymes was significantly enhanced, and the contents of lipid peroxidation products were reduced. Regarding the high dose group, it has a good inhibitory effect on oxidative damage induced by d-galactose.

A considerable volume of literature demonstrated that Ach E may lead to neuronal damage and the formation of glioma plaque in AD [43]. In the development of AD, the symptoms caused by choline dysfunction appear earlier, and neuronal apoptosis is also dominated by cholinergic neurons. Therefore, when neurons are damaged, Ach E activities increase, body Ach levels decrease, and memory is impaired. When the function of neurons is restored, the activities of Ach E decrease, the levels of acetylcholine in the body increase, and the memory is improved [44]. Studies have shown that nitric oxide (NO) is a transmitter or medium with a variety of biological activities, and has extremely important physiological and pathological effects in the body. During the aging process, the levels and function of NO decrease with the increase of age. As a key enzyme and rate-limiting enzyme in the process of NO synthesis, the change of NOS activities could reflect the levels of NO in vivo. In our experiments, the activities of NOS were markedly elevated, while the activities of Ach E were markedly depressed by feeding of polysaccharides, indicating that SFPS showed potential repairing effects on nerve cell damage caused by d-galactose.

Obviously, the biological activities of edible fungi polysaccharides are associated with their characterizations [45]. From our results, the FT-IR, NMR and HPLC spectroscopy analysis of SFPS and FPS, and the results revealed that a predominant glucosyl linkage is α form, while SFPS has a smaller molecular weight, which is in accordance with the important role of glucose in the antioxidant process. These results are consistent with previous studies on *Lentinus edodes* polysaccharides activities reported by Zhang et al. and the chemical shifts correspond to C2 and C6 [46]. However, the abnormality of the SFPS image peaks were stronger than FPS, which indicated that the hydroxyl group on the monosaccharide molecule in the macromolecular chain of the polysaccharide was replaced after the sulfate modification, thus providing a material basis for its differentiation from the biological activities of FPS. However, because of the difference of strains, culture conditions and extraction methods, there were differences in respect to the biological activities and characterization of polysaccharides.

## 4. Materials and Methods

### 4.1. Chemicals and Reagents

The fruiting body of *F. velutipes* in this experiment was obtained from local commercial markets (Taian, China). The diagnostic kits used for investigating the activities of superoxide dismutase (SOD), catalase (CAT), and glutathione peroxidase (GSH-Px) and the total antioxidant capacity (T-AOC), as well as the contents of lipid peroxide (LPO) and malondiadehyde (MDA) were purchased from Nanjing Jiancheng Bioengineering Institute (Nanjing, China). The kits for acetylcholin esterase (Ach E), and nitric oxide synthase (NOS) investigation were supplied by Jiangsu Meibiao Biological Technology Company Limited (Yancheng, China). All the other chemicals and reagents used in present work were provided by local chemical suppliers in China.

### 4.2. Preparation of FPS and SFPS

The fruiting-bodies were naturally dried and pulverized into powder using a disintegrator (Shanghai, China). The FPS or SFPS was prepared referencing to the method of Li et al. and Jia et al. with slight modifications [47,48]. The *F. velutipes* powder was extracted in distilled water (1:20, w/v) at 90 °C for 4 h. After centrifugation (4000 r/min, 15 min), the supernatant was precipitated with three volumes of ethanol (95%, v/v) overnight (4 °C). The precipitate was collected by centrifugation (3000 r/min, 15 min) and deproteinated by employing the Sevage method [49], dialyzed for removing the distilled water and lyophilized by vacuum freeze drying to obtain FPS.

The preparation of SFPS was applied by chlorosulfonic acid-pyridine method. The chlorosulfonic acid (10 mL) was dropped into a three-necked flask containing pyridine (50 mL) for 50 min to prepare an esterification reagent. After preparing the esterification reagent, the FPS (2.0 g) was dissolved in *N*, *N*-dimethylformamide solution (40 mL), thoroughly stirred and mixed with the esterification reagent in water bath (60 °C) for 3 h. After the reaction completed, it was cooled to room temperature, and sodium hydroxide solution (10%, w/v) was added to adjust until neutrality. Then, three-fold volumes of ethanol (95%, v/v) were added overnight (4 °C), and the precipitate was obtained by centrifugation (8000 r/min, 15 min). The precipitate was dissolved in water, dialyzed for 5 d, and vacuum-dried to obtain SFPS [50].

### 4.3. Degree of Substitution (DS)

The sulfur content of the sample was determined by Xu. et al. method with some modifications [51]. The 5.0 mg SFPS samples were dissolved in hydrochloric acid (3 mL, 1 mol/L) and sealed, and hydrolyzed at 100 °C for 6 h. The 0.5 mL of hydrolysate was taken out, and the samples were analyzed by barium chloride-gelatin turbidimetry. The DS was calculated from the sulfur content using the following Equation (1) according to the previous method.
DS = [(1.62 × S %)/(32 − 1.02 × S%)] × 100%(1)
where S% is the mass fraction of S atom.

### 4.4. Structural Characterization of FPS and SFPS

#### 4.4.1. Ultraviolet (UV) Analysis and Molecular Weight Determination

The purity of the FPS and SFPS were dissolved and diluted to a proper concentration, and scanned from 200 to 400 nm with the UV spectrophotometer. The carbohydrates contents (FPS and SFPS) were determined by the phenol sulfuric acid colorimetric method, using glucose as a standard. The molecular weights and homogeneities were determined by a high-performance thin layer chromatography (HPLC) system (Shimadzu LC-2010AT, Kyoto, Japan) equipped with an Atlantis C18 column (250 mm × 4.6 mm × 5 µm) and a refractive index detector. The deionized water was used as the mobile phase with a flow rate of 1 mL/min, and the column temperature was maintained at 30 °C. A series of standard dextran solutions were used to generate the calibration curve, and the molecular weights were analyzed using Agilent GPC software.

#### 4.4.2. Fourier-Transform Infrared (FT-IR) Spectroscopy Analysis

The FT-IR spectra of the FPS or SFPS were determined with a Thermo-Nicolet 6700 FT-IR spectrophotometer (Thermo Scientific, Waltham, MA, USA) during the range of 4000–400 cm^−1^. Each spectrum was obtained with 128 scans at a resolution of 2 cm^−1^ and background scans were obtained using KBr powder.

#### 4.4.3. Nuclear Magnetic Resonance (NMR) Analysis

The FPS or SFPS (50 mg) was dissolved in 2 mL of deuterated water (D_2_O). ^1^H and ^13^C NMR spectra were recorded at 25 °C by using a Bruker AV-300 spectrometer with 300 MHz.

#### 4.4.4. Monosaccharide Composition Analysis

The monosaccharide compositions were analyzed by HPLC as described by Sun. et al. with slight modifications [52]. The samples were taken into a 5 mL ampule, sealed with trifluoroacetic acid (TFA) (2 mL, 2 mol/L), and acidified at 110 °C for 8 h, evaporated with TFA, and reconstituted with 2.0 mL of water. The NaOH (250 μL, 0.6 mol/L), PMP-methanol (500 μL, 0.4 mol/L) were added to the 250 μL sample hydrolyzate, and reacted at 70 °C for 1 h. After cooling for 10 min, HCl (500 μL, 0.3 mol/L) were added for neutralization, 1 mL of chloroform was added and shaken vigorously for 1 min, centrifuged at 3000 r/min for 10 min, and the supernatant was taken and extracted three times. The supernatant was diluted once with water to obtain a sample solution derivative. The composition identification was analyzed by comparison with standard monosaccharides.

### 4.5. In Vitro Antioxidant Activities

The scavenging abilities on hydroxyl radicals were measured by the method of Smironff and Cumbes with slight modifications [53]. The sample (1 mL) was added to the phosphate buffer solution (5 mL, 0.2 mol/L, pH 7.4), then ferrous sulfate (1 mL, 7.5 mmol/L) and phenanthroline (1 mL, 7.5 mmol/L) were mixed immediately, and then the hydrogen peroxide (1 mL, 3%) was added incubated at 37 °C for 1 h. The absorbance of the mixture was measured at 560 nm and the scavenging rate on hydroxyl radicals was calculated as the following Equation (2).
Scavenging abilities (%) = (1 − A_1_/A_0_) × 100(2)
where A_1_ is the absorbance of the polysaccharide, and A_0_ is the absorbance of the blank group.

The mixture of polysaccharides sample (1 mL) and Tris/HCl buffer (4.5 mL, 50 mM, pH 8.2) were incubated at 25 °C for 20 min, then 0.3 mL pyrogallol solution (3 mM, m/v) was added at 25 °C pre-temperature for 20 min in the reaction system and mixed thoroughly. After the system was accurately reacted for 3 min, 1.0 mL of ascorbic acid (5%, m/v) was added to terminate the reaction. The absorbance was measured at 420 nm, and the scavenging superoxide radical abilities were calculated by the following Equation (3).
Scavenging abilities (%) = (1 − A_1_/A_0_) × 100(3)
where A_1_ is the absorbance of the polysaccharide samples, and A_0_ is the absorbance of the blank.

The scavenging effects on 2, 2-diphenylpicrylhydrazyl (DPPH) radicals were measured according to the method of Cumby et al. with minor modifications [54]. The 2 mL of polysaccharide solution (100–1000 mg/L) was added to 2 mL DPPH (0.2 μmol/L). After mixing evenly, the reaction was protected from light for 30 minutes. The absorbance was measured at 517 nm and the scavenging ability was calculated following Equation (4).
Scavenging abilities (%) = (1 − A_1_/A_0_) × 100(4)
where A_1_ is the absorbance of the tested samples, and A_0_ is the absorbance of the blank.

The reducing power was determined according to the procedure described with some modifications [55]. The reaction system was 1.0 mL of polysaccharide solution, adding 2.5 mL of phosphate buffer (pH 6.6, 0.2 mol/L), 1 mL of potassium ferricyanide (1%, w/v), and were incubated at 50 °C for 20 min. After cooling with running water, 2.0 mL of a trichloroacetic acid solution (10%, w/v) was added to terminate the reaction. After centrifugation at 3000 r/min for 10 min, 2.5 mL of the supernatant was taken, and 2.5 mL of deionized water and 1.2 mL of ferric chloride (0.1%, w/v) were added. After preheating at 25 °C for 15 min, the absorbance was determined at 700 nm to obtain the reducing power values directly.

### 4.6. Acute and Subchronic Toxicity Experiment

Both acute and subchronic toxicity experiments were processed using methods reported by previous literature. Fifty male Kunming strain mice were collected for the acute toxicity study. The mice were divided into four dose groups and one normal control (NC) group (total of ten groups, five females and five males). The mice in the experimental groups (SFPS/FPS) received oral polysaccharides at doses of 1000 and 3000 mg/kg/d, while the NC group received isometric saline solutions. All mice were allowed to feed for 14 consecutive days and their appearance parameters, behavioral changes or mortality were observed regularly.

The other mice were divided into six dose groups and one normal control group of ten mice per group (5 females and 5 males). The six-dose groups received SFPS/FPS of 1000, 2000 and 4000 mg/kg/d, respectively. The initial and final body weights of the mice were recorded during the entire gavage period. After the 28th day, the mice were sacrificed, and the appearance of the organs such as the heart, liver, spleen, lung and kidney were observed by the naked eye for abnormal pathological changes, and blood physiological and biochemical indexes were measured. In the hematological examination, the red blood cell count (RBC), white blood cell count (WBC), and hemoglobin levels (HB) were determined. The determination of alanine aminotransferase (ALT) activities, aspartate aminotransferase (AST) activities, alkaline phosphatase (ALP) activities, albumin levels (Ab), blood urea nitrogen levels (BUN), creatinine levels (CRE), blood glucose levels (GLU), total protein levels (TP) and triglyceride levels (TG) in serum biochemical indicators were undertaken using an automatic serum biochemical analyzer (ACE, USA).

### 4.7. Animal Experiments

The male Kunming strain mice (weighted 20 ± 2 g, 6–8 weeks) were purchased from Taibang Biological Company (Taian, China). All mice were placed in cages and adapted to one week under controlled conditions (temperature 23 ± 2 °C, lights on 12 h every day). All experiments were performed in accordance with the Regulations of Experimental Animal Administration issued by the State Committee of Science and Technology of the People’s Republic of China.

After the acclimation, all mice were randomly divided into seven groups (ten in each group) including four dosage groups, as well as one normal control (NC) group, one model control (MC) group and one positive control (PC) group. The aging mice model was successfully established by intraperitoneal injection d-galactose (300 mg/kg/d) for 56 consecutive days [56]. During the experimental procedure, the mice of the dosage groups were treated with SFPS or FPS for the high-dosage group (400 mg/kg) and the low-dosage group (200 mg/kg) dissolved in distilled water, respectively, as well as the ascorbic acid (300 mg/kg) in the PC group and the same volume of isometric saline water in the NC group. At the same time, the mice in NC group received free access to water and standard food, while the other groups were intraperitoneally injected with d-galactose (300 mg/kg), respectively. At the end of the experiment, all mice were anesthetized and sacrificed by exsanguinations after fasting overnight.

The liver, kidney and brain were separated from the body and the organ weight were weighed to calculate the organ index coefficients. The animal organ index calculation method was defined as the coefficient (mg/g) = organ weight (mg)/body weight (g).

The blood sample was obtained from retrobulbar vein, and the serum was obtained after centrifugation (14,000 r/min) at 4 °C for 10 min. The activities of serum alanine aminotransferase (ALT), aspartate aminotransferase (AST), alkaline phosphatase (ALP), urea nitrogen (BUN), creatinine (CRE), and uric acid (UA), high density lipoprotein cholesterol (HDL-C) and low-density lipoprotein cholesterol (LDL-C) were measured using the automatic biochemical analyzer (ACE, USA).

The liver, kidney or brain were freshly harvested, weighted and immediately homogenized (1:9, w/v) in phosphate buffer (0.2 M, pH 7.4, 4 °C), and then centrifuged at 5000 r/min for 20 min (4 °C). The supernatants were stored at −20 °C for further biochemical analysis. The activities of SOD, GSH-Px, CAT and T-AOC, as well as contents of LPO and MDA were assayed using commercially available diagnostic kits (Nanjing Jiancheng Bioengineering Institute, Nanjing, China).

The levels of Ach E and NOS were determined by commercial reagent kits following the manufacturer’s instructions (Jiangsu Meibiao Biological Technology Company Limited, Haimen, China). The optical density of each well was assayed at 412 nm, 530 nm with a microplate spectrophotometer. Finally, the contents were calculated according to the standard curve.

The fresh liver or brain pathological tissues were preserved in 4% formalin, embedded in paraffin, and then cut into 5 μm thick sections. The histopathological analysis was performed under a light microscope (×400 magnifications) after hematoxylin and eosin staining.

### 4.8. Statistical Analysis

All the data were expressed as the mean ± S.D. (standard deviation). The statistical significance of the differences among groups was analyzed by ANOVA in the SPSS 19.0 statistical package from International Business Machines Corporation (USA). The *p* < 0.05 was considered as statistically significant.

## 5. Conclusions

The present work showed that the SFPS had potential antioxidant abilities in vitro, anti-aging and protective effects in vivo, against the d-galactose-induced aging mice. Furthermore, the FT-IR and NMR structural characterization analysis indicated that the β-glucosyl linkage was present in SFPS. The UV and HPLC analysis showed that the SFPS were homogeneous compounds with an average molecular weight (Mw) of 2.81 × 10^3^ Da. These results demonstrated that the SFPS could be used as a natural resource for the development of potential functional foods and the prevention of aging and its complications.

## Figures and Tables

**Figure 1 molecules-24-03517-f001:**
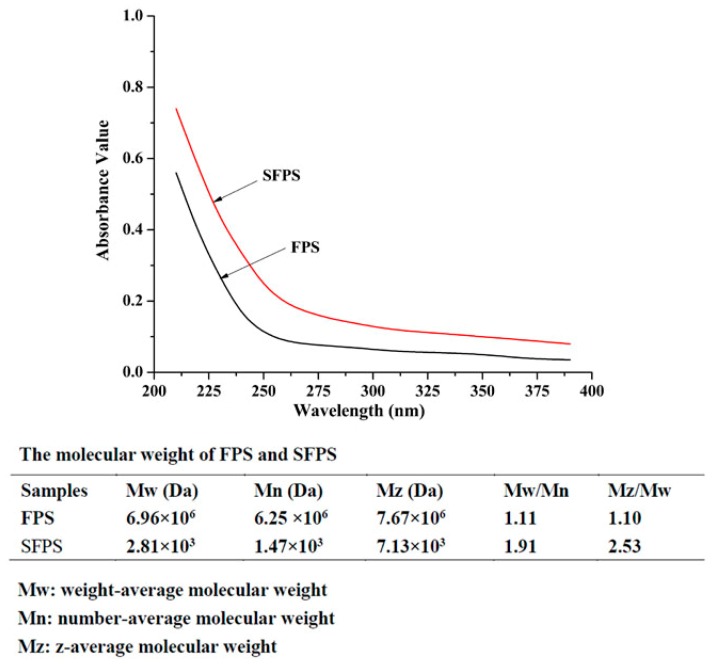
The purity, and molecular weight of sulfated *Flammulina velutipes* polysaccharides (SFPS), *Flammulina velutipes* polysaccharides (FPS).

**Figure 2 molecules-24-03517-f002:**
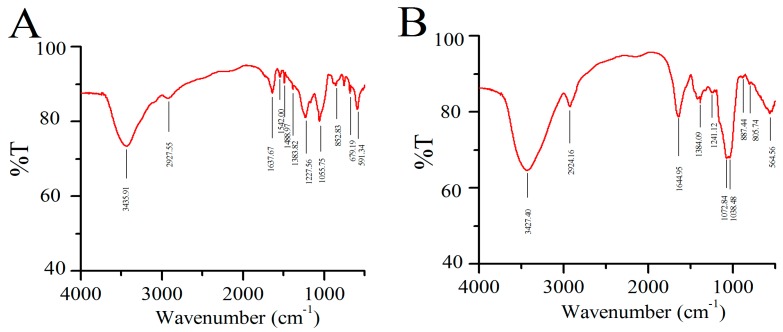
FTIR spectra. (**A**) FPS, (**B**) SFPS.

**Figure 3 molecules-24-03517-f003:**
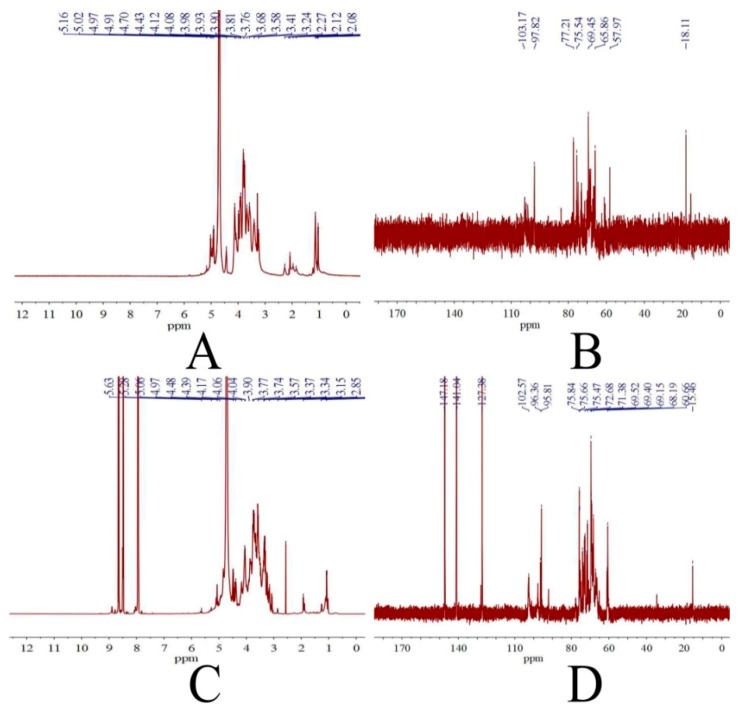
NMR spectra. ^1^H NMR spectra of FPS (**A**), SFPS (**C**) and ^13^C. NMR spectra of FPS (**B**), SFPS (**D**).

**Figure 4 molecules-24-03517-f004:**
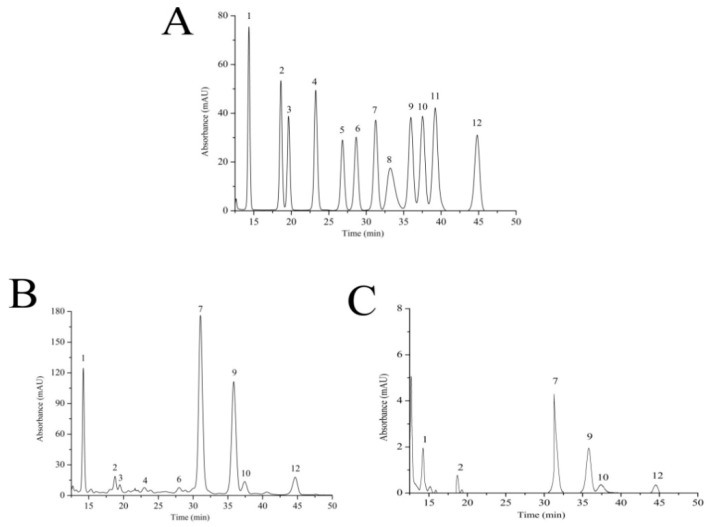
High performance liquid chromatography of (**A**) 12 standard monosaccharides, (**B**) FPS and (**C**) SFPS. Peaks: (1) mannose, (2) ribose, (3) rhamnose, (4) glucuronic acid, (5) galacturonic acid, (6) glucosamine, (7) glucose, (8) amino-galactose, (9) galactose, (10) xylose, (11) arabinose and (12) fucose.

**Figure 5 molecules-24-03517-f005:**
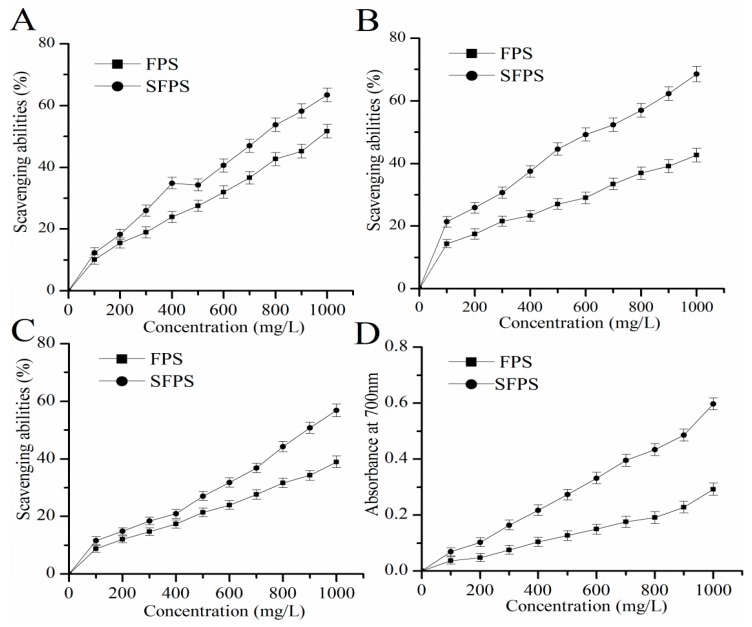
The in vitro antioxidant activities of SFPS and FPS. (**A**) hydroxyl radicals, (**B**) Superoxide radicals, (**C**) DPPH radicals and (**D**) reducing power.

**Figure 6 molecules-24-03517-f006:**
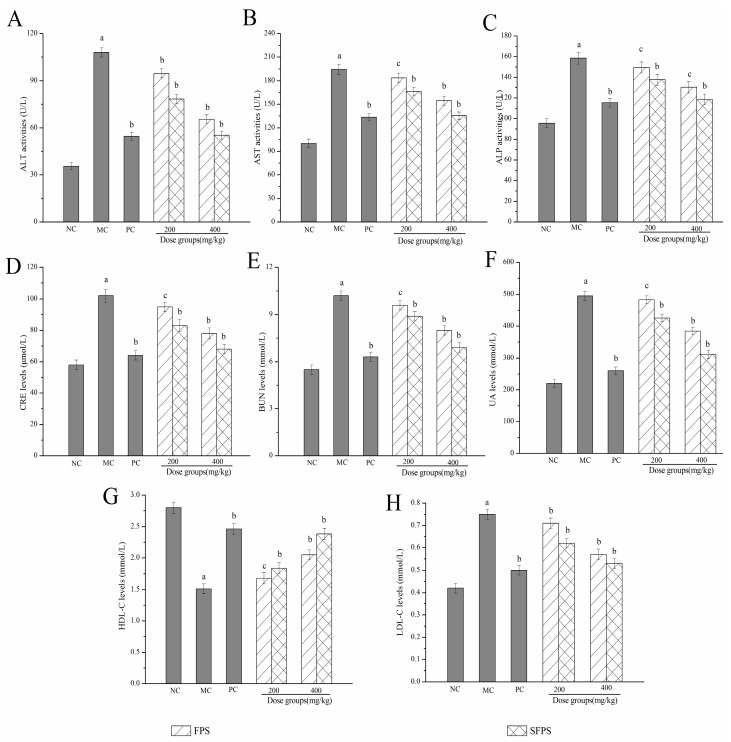
Effects of SFPS and FPS on (**A**) alanine aminotransferase (ALT) activities, (**B**) aspartate aminotransferase (AST) activities, (**C**) alkaline phosphatase (ALP) activities (**D**) creatinine levels (CRE) levels (**E**) blood urea nitrogen levels (BUN) levels (**F**) uric acid (UA) levels (**G**) high density lipoprotein cholesterol (HDL-C) levels and (**H**) low density lipoprotein cholesterol (LDL-C) levels in serum. The values were expressed as the means ± S.D. of ten mice per group. (a) *p* < 0.01 compared with NC group, (b) *p* < 0.01, and (c) *p* < 0.05 compared with the MC group.

**Figure 7 molecules-24-03517-f007:**
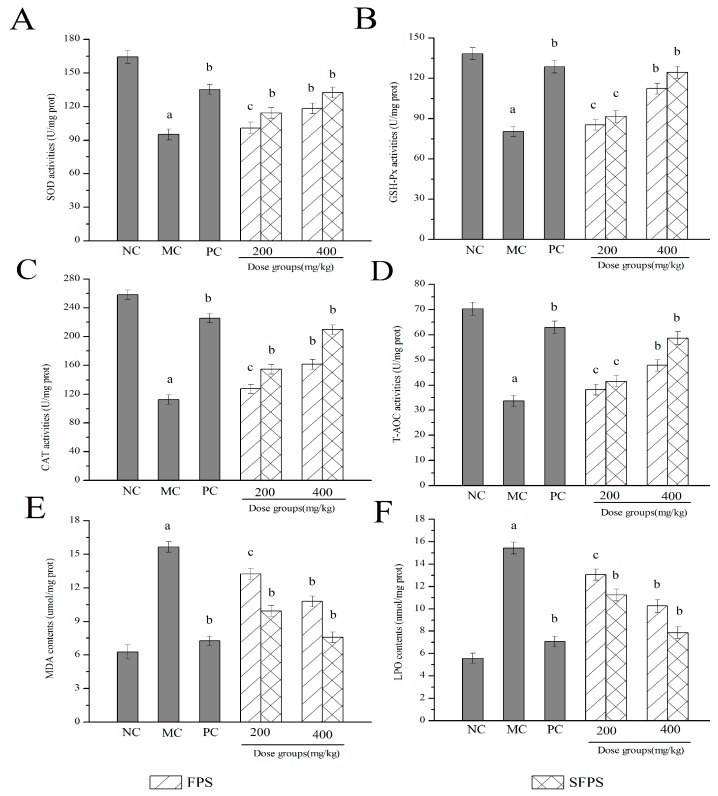
Effects of SFPS and FPS on the antioxidant enzymes of (**A**) superoxide dismutase (SOD), (**B**) glutathione peroxidase (GSH-Px), (**C**) catalase (CAT), and (**D**) the total antioxidant capacity (T-AOC), as well as contents of (**E**) malondiadehyde (MDA) and (**F**) lipid peroxide (LPO). The values were expressed as the mean ± S.D. of ten mice per group, (a) *p* < 0.01 compared with NC group, (b) *p* < 0.01 compared with MC group, (c) *p* < 0.05 compared with MC group.

**Figure 8 molecules-24-03517-f008:**
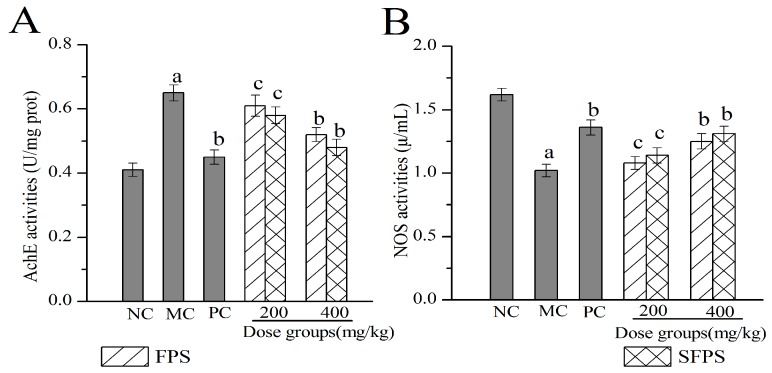
Effects of SFPS and FPS on (**A**) acetylcholin esterase (Ach E) and (**B**) nitric oxide synthase (NOS) in the brain. The values were expressed as the mean ± S.D. of ten mice per group, (a) *p* < 0.01 compared with NC group, (b) *p* < 0.01 compared with MC group, (c) *p* < 0.05 compared with MC group.

**Figure 9 molecules-24-03517-f009:**
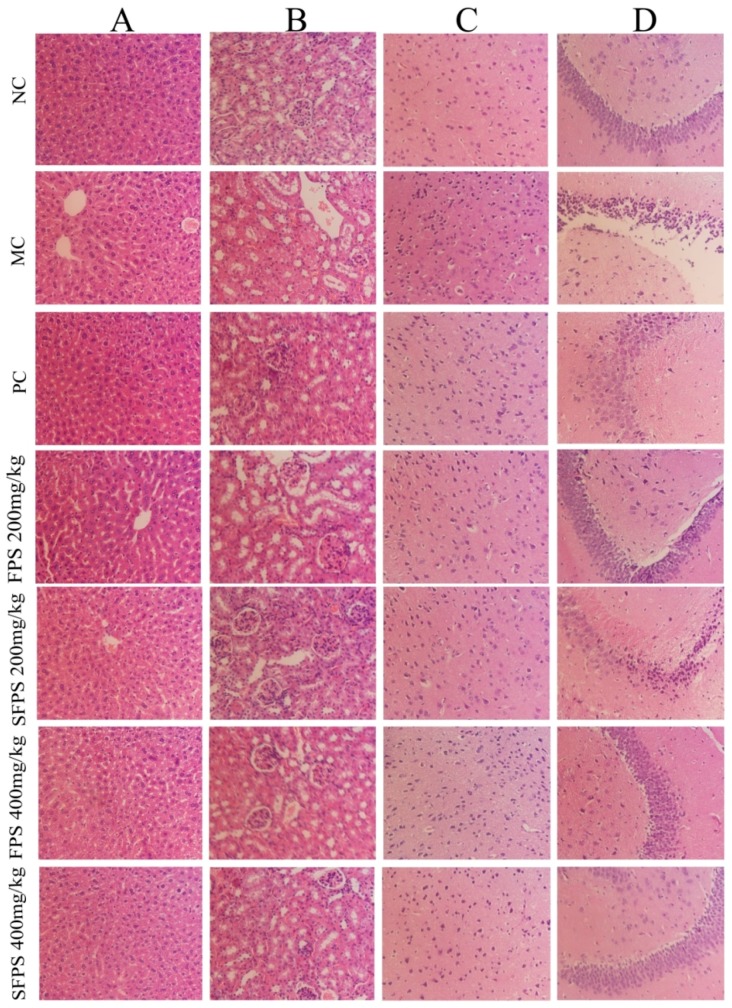
Effects of SFPS and FPS on organ damages of (**A**) liver, (**B**) kidney, (**C**) the cortex of mice brain and (**D**) the hippocampus of mice brain with HE staining (magnification 400×).

**Table 1 molecules-24-03517-t001:** Effects of SFPS and FPS on body weight and organ index in d-gal-treated mice.

Groups	Body Weight (g)	Organ Index (mg/g)		
		Liver Index	kidney Index	Brain Index
NC	45.91 ± 2.46	5.99 ± 0.46	4.61 ± 0.43	1.25 ± 0.08
MC	37.86 ± 2.28 ^a^	4.53 ± 0.41 ^a^	4.09 ± 0.35 ^a^	1.02 ± 0.05 ^a^
PC	42.78 ± 3.02 ^b^	5.38 ± 0.38 ^b^	4.58 ± 0.42 ^b^	1.21 ± 0.04 ^b^
FPS				
200 mg/kg/d	38.78 ± 2.32 ^c^	4.70 ± 0.39 ^c^	4.12 ± 0.36 ^c^	1.08 ± 0.04 ^c^
400 mg/kg/d	40.94 ± 2.31 ^c^	5.07 ± 0.45 ^c^	4.36 ± 0.40 ^b^	1.15 ± 0.05 ^c^
SFPS				
200 mg/kg/d	39.86 ± 2.30 ^c^	4.84 ± 0.36 ^c^	4.26 ± 0.34 ^c^	1.11 ± 0.04 ^c^
400 mg/kg/d	42.55 ± 3.05 ^b^	5.26 ± 0.37 ^b^	4.52 ± 0.44 ^b^	1.19 ± 0.07 ^b^

The values were reported as the Mean ± S.D. of ten mice in each group. NC: normal control group, MC: model control group and PC: positive control group. (a) *p* < 0.01, compared with NC groups; (b) *p* < 0.01, compared with MC groups; (c) *p* < 0.05, compared with MC groups.

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
