# Peer review of "Characterization, Antioxidant, Anti-Aging and Organ Protective Effects of Sulfated Polysaccharides from Flammulina velutipes"

_molecules, 2019, doi:10.3390/molecules24193517_

Round 1

Reviewer 1 Report

Dear Authors,

I would like to see your studies of the biocompatibily ( at different concentrations and treatment times) of sulfated polysaccharides from Flammulina velutipes. This aspects are very important for the safety in clinical applications.

Author Response

I would like to see your studies of the biocompatibily ( at different concentrations and treatment times) of sulfated polysaccharides from Flammulina velutipes. This aspects are very important for the safety in clinical applications.

Thanks a lot for your valuable suggestions. In the early stage, we did a lot of toxicity tests with polysaccharides. The section of toxicity experiment was written and marked in red in the manuscript.

4.6. Acute and subchronic toxicity experiment

Both acute and subchronic toxicity experiments were processed using methods reported by previous literatures. Fifty male Kunming strain mice were collected for the acute toxicity study. The mice were divided into four dose groups and one normal control (NC) group (Total ten groups, five females and five males). The mice in the experimental groups (SFPS/FPS) received oral polysaccharides at doses of 1000 and 3000 mg/kg/d, while the NC group received isometric saline solutions. All mice were allowed to feed for 14 consecutive days and their appearance parameters, behavioral changes or mortality were observed regularly.

The other mice were divided into six dose groups and one normal control group of ten mice per group (5 females and 5 males), the sixe-dose groups received SFPS/FPS of 1000, 2000 and 4000 mg/kg/d, respectively. The initial and final body weights of the mice were recorded during the entire gavage period. After the 28th day, the mice were sacrificed, and the appearance of the organs such as heart, liver, spleen, lung and kidney were observed by the naked eye for abnormal pathological changes, and blood physiological and biochemical indexes were measured. In the hematological examination, the red blood cell count (RBC), white blood cell count (WBC), and hemoglobin levels (HB) were determined. Determination of alanine aminotransferase (ALT) activities, aspartate aminotransferase (AST) activities, alkaline phosphatase (ALP) activities, albumin levels (Ab), blood urea nitrogen levels (BUN), , creatinine levels (CRE), blood glucose levels (GLU), total protein levels (TP) and triglyceride levels (TG) in serum biochemical indicators using automatic serum biochemical analyzer (ACE, USA).

2.3. Acute and subchronic toxicity analysis

The mice in each group were continuously observed for 14 days, and there were no abnormalities in their eating, drinking, activity, mental state, smooth fur, normal feces, and no death. There was no abnormality in the mental state, activity status, drinking water, and fecal urine of the mice in each dose group. The mice in the 6 dose groups (SFPS/FPS) were not significantly different from the normal control (NC) group in weight, hematological parameters and serum biochemical indexes after 28 days of continuous gavage, and male mice showed superior tolerance. The results demonstrated that the polysaccharides of SFPS and FPS showed superior biocompatibilies.

Reviewer 2 Report

The research article that is authored by Fangfang Yuan et al, describes the potential antioxidant abilities of Flammulina velutipes polysacharids and its sulfated form in-vitro, and its antiaging and protective effects in-vivo. I suggest accepting the manuscript for publication in Molecules journal subject to revisions as described below:

Page 4, lines 115-116, “The galactose contents in SFPS was agreement with…..” should be corrected. Page 6, Table 1: What do you mean by groups MC / PC? Page 7, line 152: The authors wrote (P<0.05 or <0.01). Please clarify? What is the real P-value? Page 7, lines 151-154: the comparison is between results of SFPS at dosage of 400 mg/kg and 400 mg/kg. Something is missing or wrong there. Page 11, line 241: The authors talk about mushroom polysaccharides while citing reference no. 29 , which describe the biological effects of pomegranate seed oil.

Author Response

The research article that is authored by Fangfang Yuan et al, describes the potential antioxidant abilities of Flammulina velutipes polysacharids and its sulfated form in-vitro, and its antiaging and protective effects in-vivo. I suggest accepting the manuscript for publication in Molecules journal subject to revisions as described below:

1 Page 4, lines 115-116, “The galactose contents in SFPS was agreement with…..” should be corrected.

Thanks for your kind suggestion. “The galactose contents in SFPS was agreement with…..” has been changed to “The content of galactose in SFPS was agreement with the antioxidant activities, indicating galactose may play vital roles in maintaining the antioxidant status”.

2 Page 6, Table 1: What do you mean by groups MC / PC?

Thanks for your useful suggestion. The meaning of MC,PC and NC has been marked in the manuscript. MC: model control group, PC: positive control group and NC: normal control group.

3 Page 7, line 152: The authors wrote (P<0.05 or <0.01). Please clarify? What is the real P-value? Page 7, lines 151-154: the comparison is between results of SFPS at dosage of 400 mg/kg and 400 mg/kg. Something is missing or wrong there.

Thanks for your valuable comments. Since the description was not clear before, it has been corrected as follows: Fortunately, these enhancements could be inhibited by the treatment with SFPS (P < 0.05) or FPS (P < 0.05). Especially SFPS at the dosage of 400 mg/kg, the ALT, AST and ALP activities were decreased to 55.23 ±3.57, 135.34 ± 4.90 and 118.43 ± 5.11 U/L, respectively.

4 Page 11, line 241: The authors talk about mushroom polysaccharides while citing reference no. 29 , which describe the biological effects of pomegranate seed oil.

Thanks for your professional advice. References 26 has been changed to “Kou, L.; Du, M.Z.; Liu, P.J.; Zhang, B.H.; Zhang, Y.Z.; Yang, P.; Shang, M.Y.; Wang, X.D. Anti-Diabetic and Anti-Nephritic Activities of Grifola frondosa Mycelium Polysaccharides in Diet-Streptozotocin-Induced Diabetic Rats Via Modulation on Oxidative Stress. Appl. Biochem. Biotechnol. 2019, 187, 310-322”.

Reviewer 3 Report

It is a very interesting article. This paper attempts to prove the antioxidant and anti-aging potential and organ protective effects of polysaccharides (FPS) and its sulfated FPS (SFPS) extracted from the edible mushroom specie Flammulina velutipes.

Overall, the methodology used could be improved. The manuscript is very well organized and written and perfectly feet the journal’s scope. It’s very well written and supported overall by recent bibliographic references.

Comments or suggestions:

Title is concise and detailed enough to understand the paper content. Flammulina velutipes must be written in italics. Authors should correct it in all document.

The abstract summarises the content of the article. It’s brief and clear. But in my opinion, the abstract should start with an introductory sentence.

Line 11:”The characteristics,..” Authors should clarify which characteristics.

Line 13: “in vitro”. Italics

Line 14: “in vivo”. Italics

The keywords repeat the words of the title. Avoid using the same title words as this will increase the probability that the article will be detected in a search. So, I suggest that authors correct the keywords.

Introduction highlights all the subjects needed to understand all the manuscript. It's clear and detailed.

Results data is organized in an orderly and logical sequence.

The most relevant results are highlighted.

The discussion explains the data presented in the results, mostly compared with recent work.

Materials and Methods are very well described.

Line 444: “The activities of SOD, GSH-444 Px, CAT and T-AOC, as well as contents of LPO and MDA were determined by the commercial reagent kits according to the instructions. The levels of Ach E and NOS were determined with ELISA kits according to the relevant kits.” Please identify the kits used.

Author Response

Comments or suggestions:

1 Title is concise and detailed enough to understand the paper content. Flammulina velutipes must be written in italics. Authors should correct it in all document.

Thanks a lot for your technical comments. All Flammulina velutipes have been italicized in the manuscript.

2 The abstract summarises the content of the article. It’s brief and clear. But in my opinion, the abstract should start with an introductory sentence.

Thanks a lot for your meaningful comments. The abstract has been introduced “Aging is the cause of many diseases. However, polysaccharide plays an important role in antioxidant and anti-aging. In this paper, the antioxidant and anti-aging activities of Flammulina velutipes polysaccharides (FPS) and its sulfated FPS (SFPS) on d-galactose-induced aging mice were investigated.” in the manuscript.

3 Line 11:”The characteristics,..” Authors should clarify which characteristics.

Thanks for your comments. This sentence has been deleted.

4 Line 13: “in vitro”. Italics; Line 14: “in vivo” Italics.

Thanks for your useful suggestion. The two word groups have been transformed into “in vitro” and “in vivo”, respectively.

5 The keywords repeat the words of the title. Avoid using the same title words as this will increase the probability that the article will be detected in a search. So, I suggest that authors correct the keywords.

Thanks for your appreciation and affirmation of our manuscript. The keywords have been changed to “Sulfated polysaccharides; Flammulina velutipes; Antioxidation; Anti-aging effects; Organ protection”.

6 Line 444: “The activities of SOD, GSH-444 Px, CAT and T-AOC, as well as contents of LPO and MDA were determined by the commercial reagent kits according to the instructions. The levels of Ach E and NOS were determined with ELISA kits according to the relevant kits.” Please identify the kits used.

Thanks for your professional advice. This sentence has been corrected as “The activities of SOD, GSH-Px, CAT and T-AOC, as well as contents of LPO and MDA were assayed using commercially available diagnostic kits (Nanjing Jiancheng Bioengineering Institute, Nanjing, China). The levels of Ach E and NOS were determined by commercial reagent kits following the manufacturer’s instructions (Jiangsu Meibiao Biological Technology Company Limited, Jiangsu, China).” in the manuscript.

Reviewer 4 Report

The manuscript entitled "Characterization, antioxidant, anti-aging and organ protective effects of sulfated polysaccharides from Flammulina velutipes" deals with the characteristics of the polysaccharides and the sulfated polysaccharides of the edible mushroom enoki, focusing on their antioxidant properties and their ability to slow down the aging process induced by d-galactose in mice.

The study is scientifically sound and deserves publication in "Molecules". However, I am not convinced that there are practical applications of these findings. Can the authors explain why FPS and SFPS may be preferred over ascorbic acid, since in all experiments presented in the manuscript ascorbic acid (PC) is equally potent or even more active than FPS / SFPS ?

A paragraph regarding the d-galactose-induced aging mouse model is necessary in the Introduction (probably the authors can use the information provided in the beginning of the Discussion; page 11, lines 227-239).

The term "organ index" should be defined in Materials and Methods, while Table 1 should be corrected, because now body weight seems to be a subset of organ index.

The author have to check for some minor language or spelling mistakes, e.g. in page 12 in line 247 should read "significant" instead of "significative" and in line 254 "The" should be written as "the".

Author Response

Reviewer 4 comments:

1 The study is scientifically sound and deserves publication in "Molecules". However, I am not convinced that there are practical applications of these findings. Can the authors explain why FPS and SFPS may be preferred over ascorbic acid, since in all experiments presented in the manuscript ascorbic acid (PC) is equally potent or even more active than FPS / SFPS ?

Thanks a lot for your meaningful comments. As a macromolecular polymer of non-toxic and non-side-effects substances, the edible mushroom polysaccharide could be decomposed into a variety of short-chain fatty acids by the intestinal flora in the body, further exerts physiological functions. While ascorbic acid is a simple product with a simple structure. And the long-time use of ascorbic acid may cause side effects including polyuria, diarrhea and skin rash, etc.. When taking ascorbic acid for a long time, it is easy to make urine acidic, and some alkaline substances combined with the formation of urinary calculi. Therefore, polysaccharides have more extensive and practical application value. The related references were listed as follows.

(1) M. Lauverjat, S. Ait, C. Bergoin, D. Barnoud, C. Peraldi, C. Chambrier. Vitamin C status in long-term parenteral nutrition patients. Clinical Nutrition, 2018, 37, S186-S186.

(2) Mohammadreza Sabri, Ghafour Ghaffari, Mahin Hashemipour, Neda Mostofizadeh, Ali Mehrabi Koushki. Effect of long-term Vitamin C intake on vascular endothelial function in diabetic children and adolescents: A pilot study. Journal of Research in Medical Sciences, 2016, 21(1), 119-119.

2 A paragraph regarding the d-galactose-induced aging mouse model is necessary in the Introduction (probably the authors can use the information provided in the beginning of the Discussion; page 11, lines 227-239).

Thanks for your valuable comments. An introduction to the mouse model of d-galactose-induced aging has been introduced in the introduction and is highlighted in red in the manuscript.

3 The term "organ index" should be defined in Materials and Methods, while Table 1 should be corrected, because now body weight seems to be a subset of organ index.

Thanks for your useful comments. The organ index has been defined in materials and methods as follows: The liver, kidney and brain were separated from the body and then weighed to calculate the organ index coefficients. The animal organ index calculation method is defined as the coefficient (mg/g) = organ weight (mg)/body weight (g).

4 The author have to check for some minor language or spelling mistakes, e.g. in page 12 in line 247 should read "significant" instead of "significative" and in line 254 "The" should be written as "the".

Thanks for your professional advice. The “significative” has been corrected to “significant” and “The” was also corrected to “the”.

Round 2

Reviewer 1 Report

Dear Editor,

Now I consider this manuscript clear and consistent.  I believe the manuscript is of interest to the readers of the journal and I accept this paper in this revised version.

Best regards

Annarita Stringaro